# Update of Pediatric Lipomatous Lesions: A Clinicopathological, Immunohistochemical and Molecular Overview

**DOI:** 10.3390/jcm11071938

**Published:** 2022-03-31

**Authors:** Eline Ameloot, Fleur Cordier, Jo Van Dorpe, David Creytens

**Affiliations:** 1Department of Pathology, Ghent University Hospital, Ghent University, 9000 Ghent, Belgium; eline.ameloot@uzgent.be (E.A.); fleur.cordier@uzgent.be (F.C.); jo.vandorpe@uzgent.be (J.V.D.); 2Cancer Research Institute Ghent (CRIG), Ghent University Hospital, Ghent University, 9000 Ghent, Belgium

**Keywords:** adipocytic, lipoblastoma, lipoblastomatosis, lipomatosis, liposarcoma, pediatric

## Abstract

Lipomatous neoplasms are a rare entity in the pediatric population, comprising less than 10% of soft tissue tumors in the first two decades of life. Some characteristics of pediatric adipocytic tumors are analogous to their adult counterparts, some pediatric lipomatous lesions however harbor unique features. In recent years, there have been significant advances in the understanding of the pathogenesis and hence in the classification and treatment of pediatric adipocytic tumors. This literature-based article will provide a review of the presently known clinicopathological, immunohistochemical and molecular features of pediatric lipomatous lesions.

## 1. Introduction

Although adipocytic neoplasms are the most common soft tissue tumors in adults, they are relatively rare in the pediatric population. Lipomatous tumors comprise less than 10% of soft tissue tumors in the first two decades of life [1]. Certain features of pediatric adipocytic lesions are identical to the respective lesions in adults, there are however also distinct differences, especially with respect to the distribution spectrum of the (sub)types. The clinicopathological, immunohistochemical and molecular features (see also Table 1) of pediatric lipoblastoma/lipoblastomatosis, lipomatosis and liposarcomas (well-differentiated, dedifferentiated, myxoid, pleomorphic and myxoid pleomorphic) will be reviewed in this article.

## 2. Lipoblastoma and Lipoblastomatosis

Lipoblastoma is a benign neoplasm composed of embryonal white fat, occurring predominantly in infancy and early childhood, and rarely in adulthood. The vast majority of cases (75% to 90%) are seen before the age of three years, with a slightly higher incidence in males [1,2,3,4,5,6,7,8,9,10,11]. These lesions may present as a superficial, circumscribed and localized nodule (lipoblastoma). Or it may encompass a deeper lesion in the soft tissues, with diffuse growth and infiltrative borders (lipoblastomatosis). The distinction between circumscribed and infiltrative forms is not clinically relevant, since both types recur [10]. The trunk and extremities are most often involved [2,3,4,10,12]. Lipoblastoma may also arise in the abdomen, mesentery, retroperitoneum, pelvis, inguinoscrotal or labial region, perineum, mediastinum and head/neck region [2,11,13,14]. Lung, liver, heart, colon and parotid gland lipoblastomas have also been described [2,13,14,15,16,17,18,19,20]. Colonic and mesenteric cases have been associated with intussusception and volvulus [21,22].

Lipoblastoma can compress adjacent structures and interfere with function, particularly in large abdominal tumors, mediastinal tumors, or cervical tumors [4,23,24,25]. Truncal tumors may infiltrate into the thoracic cavity, or into the spine with neuroforaminal or intraspinal invasion [2,26,27,28,29].

T1-weighted magnetic-resonance images reveal a nodular mass with intensity similar to (or lower than) that of lipoma or subcutaneous fat [2,30,31]. These lesions are typically 2–5 cm in diameter, although they can exceed 10 cm [2,3,4,7,32]. Macroscopically, lipoblastoma is a soft, lobulated, yellow, white, or tan mass. It may display myxoid nodules, cystic spaces or adipose nodules separated by fine white fibrous septae on the cut surface [10,33].

Microscopically, lipoblastoma consists of lobulated sheets of adipocytes with a spectrum of maturation, ranging from primitive stellate or spindled mesenchymal cells (myxoid areas), to multivacuolated or small, signet-ring lipoblasts, to mature adipocytes separated by fibrovascular septae (see Figure 1) [3,4,7,32,33,34]. Mast cells are common. Hyperchromasia and mild cytonuclear atypia can be observed, mitoses however are very rare and atypical mitoses are absent [3,10,33,35]. Other features, such as chondroid metaplasia, extramedullary hematopoiesis, chronic inflammation and sparsely multinucleated or floret cells may be observed [4,7,36].

The adipocytes show immunoreactivity for S100, CD56 and CD34 [4,27,37]. Primitive mesenchymal cells can show immunohistochemical staining for desmin [3]. Approximately 80% of lipoblastoma cases show aberrant PLAG1 immunohistochemical expression. Lopez-Nunez et al. however found a concordant molecular *PLAG1* rearrangement ranging from 52–60%, thus illustrating the limits of immunohistochemical staining for PLAG1 [38]. On the other hand, Warren et al. concluded that PLAG1 immunohistochemical staining is rapid and inexpensive, in contrast to molecular genetic analysis, and in their opinion, it can thus be considered a first-line diagnostic method. Furthermore, Warren et al. found that even with molecular methodologies, *PLAG1* fusions are commonly cryptic and that commercially available targeted RNA sequencing panels may not cover newly emerged fusion partners, whereas PLAG1 staining would likely be positive regardless of the *PLAG1* fusion partner [37]. Lipoblastoma typically lacks expression of p16 immunohistochemistry. This should however be used with caution, as some lipoblastomas show expression for p16 and some liposarcomas lack p16 expression, presenting a possible diagnostic pitfall [39].

In approximately 60% of the cases a simple, pseudodiploid or hyperdiploid karyotype can be found, featuring a structural alteration of 8q11–q13, leading to a rearrangement of *PLAG1* (see also Table 1) [4,38,39,40,41,42,43]. The most frequent numerical change is one or more extra copies of chromosome 8, with or without concurrent rearrangement of 8q11–q13 [44,45]. A subset of patients diagnosed with lipoblastoma have developmental delays or abnormalities, seizures, or familial lipomas. These are potentially related to larger chromosome 8q alterations that include the *PLAG1* gene [4,7,46]. *PLAG1* fusion gene partners described in lipoblastoma include *HAS2* (8q24.13), *COL1A2* (7q21.3), *RAD51B* (14q24.1), *COL3A1* (2q32.2), *RAB2A* (8q12.1–q12.2), *BOC* (3q13.2), *DDX6*, *KLF10*, *KANSL1L*, *ZEB2* and *EF1A1* [38,47,48,49,50,51,52,53,54,55]. *HMGA2* alterations are less common [38,42].

Lipoblastoma has an excellent prognosis after excision, with no risk of metastasis [1,2,3,4,6,8,10,12,32,56]. There is, however, a recurrence rate of 13–46% due to incomplete excision [6]. Recurrence has been reported as late as six years after primary excision, highlighting the need for long-term follow-up after primary excision [57].

## 3. Lipomatosis

Lipomatosis is a broad term for an overgrowth of mature adipose tissue in a wide spectrum of clinical contexts. It can occur in hamartomatous arrangements with other mesenchymal tissue types.

Most lipomatoses are infiltrative, although they can occasionally be lobulated and circumscribed. Macroscopically, the cut surfaces show yellow fat intermixed with variable amounts of other mesenchymal tissue, depending on the location. *PTEN*, *PIK3CA* and *TSC* are part of the PI3K/PTEN/AKT/TSC/mTORC1 pathway, and germline, mosaic or somatic mutations of these genes are responsible for the different lipomatosis entities described below (see Table 1). Pharmacologic inhibitors of genes in the PI3K/PTEN/AKT/TSC/mTORC1 pathway may be useful in preventing disease recurrence or progression [58,59]. Other medical therapies including sirolimus and Alpelisib/BYL719 have shown success in decreasing or stabilizing overgrowth [10,60].

Morbidity and mortality are related to mass effect or infiltration of vital structures and organs. In congenital infiltrating lipomatosis of the face (CILF), diffuse lipomatosis (DL), encephalocraniocutaneous lipomatosis (ECCL), PTEN hamartoma of soft tissue (PHOST) and PIK3CA-related overgrowth spectrum (PROS), continued growth and/or recurrence after incomplete resection is common and can lead to disrupted growth, malformation and malfunction, sometimes requiring multiple surgeries in [10,58,60]. Some lipomatosis entities (DL, ECCL and PHOST) may be the initial evidence of an associated tumor predisposition syndrome [10].

Lipomatosis entities can be divided into pediatric and acquired/adult onset types, each with different entities. Pediatric lipomatosis entities are highlighted below.

### 3.1. Congenital Infiltrating Lipomatosis of the Face (CILF)

Congenital infiltrating lipomatosis of the face (CILF) presents at birth or in early childhood as a unilateral facial swelling with hyperplasia of the underlying bone. Hyperplasia of teeth, tongue, brain and parotid gland have also been described [58,61,62]. These lesions have been associated with *PIK3CA* mutations, suggesting CILF should be included in PROS (see below) [63].

### 3.2. Diffuse Lipomatosis (DL)

Diffuse lipomatosis (DL) presents as a large, rapidly growing mass mostly involving the trunk or extremities in a segmental fashion. It has been described in association with the tuberous sclerosis complex [64,65,66].

### 3.3. Encephalocraniocutaneous Lipomatosis (ECCL)

Encephalocraniocutaneous lipomatosis (ECCL) is a sporadic neurocutaneous disorder involving tissue derived from the neural crest with ocular, cutaneous, and central nervous system abnormalities. These aberrations include ocular choristomas, colobomas, aniridia, nevus psiloliparus, lipomatosis, alopecia, intracranial and spinal lipomas, midline pilocytic astrocytoma, dysembryoplastic neuroepithelial tumor, developmental delay, and seizures. These lesions are usually found in the soft tissues of the scalp, paravertebral soft tissues, or extremities [67,68,69,70]. Mosaic *FGFR1* activating mutations have been implicated in cases [67].

### 3.4. Michelin Tire Baby Syndrome (MTBS)

Michelin tire baby syndrome (MTBS) encompasses a rare familial genodermatosis, characterized by symmetrical excess skin folds that are present at birth and resolve with age. It diffusely involves the skin in a symmetric fashion. Other features, such as facial dysmorphism (upslanting palpebral fissures, hypertelorism, wide and/or depressed nasal bridge, epicanthic folds, auricular malformations), cleft palate, genital anomalies, developmental delay, ureterocele, smooth muscle hamartoma and nevus lipomatosis may be associated [71].

### 3.5. PTEN Hamartoma of Soft Tissue (PHOST)

*PTEN* inactivating mutations are central to the development of PTEN hamartoma of soft tissue (PHOST). Many PHOST patients have Cowden syndrome or Bannayan–Riley–Ruvalcaba syndrome [72]. Patients typically present with an intramuscular mass, as well as pain and swelling of the affected limb. The lower extremities are most frequently involved, followed by the upper extremities, trunk, head and neck [10]. Histologically, these lesions show a distinctive multinodular combination of mature adipose tissue, fibrous tissue, a vascular component, lymphoid follicles, bone and hypertrophic nerves with “onion bulb” features [72].

### 3.6. PIK3CA-Related Overgrowth Spectrum (PROS)

PIK3CA-related overgrowth spectrum (PROS) presents as asymmetric and disproportionate overgrowth lesions, with enlargement of the affected region [10]. These lesions usually present in later childhood or early adulthood, more frequently affecting the lower extremities than the upper extremities [33,73]. They can present unilaterally and can be static or progressive [10]. PROS is associated with general adipose dysregulation and can show a marked absence of adipose tissue in unaffected limbs [10,73,74]. Different clinical manifestations of this overgrowth occur in various patterns, including fibroadipose overgrowth (FAO); congenital lipomatous overgrowth/vascular malformations/epidermal nevi, scoliosis/skeletal and spinal syndrome (CLOVES), macrodactyly, megalencephaly-capillary malformation syndrome (MCAP) and hemihyperplasia multiple lipomatosis (HMML). *PIK3CA* mutations are identified in PROS, with a correlation between specific mutations (genotype) and their physical manifestations (phenotype) [73,75].

### 3.7. Nevus Lipomatosis Superficialis (NLS)

Nevus lipomatosis superficialis (NLS) typically presents as one or more soft, non-tender papules, nodules, or pedunculated, exophytic lesions in later childhood or early adulthood [10]. The mature adipose tissue is usually located in the superficial dermis, and surgical excision is usually adequate, with rare recurrences [33,76].

### 3.8. Lipomatosis of Nerve (LN)

Lipomatosis of Nerve (LN) usually presents in later childhood or early adulthood, although lesions can present at birth [3]. The lesions most frequently affect the median nerve, followed by the ulnar and plantar nerve. Histologically, these lesions show characteristic nerve expansion with perineural thickening/hypertrophy, similar to perineurioma [3,10,63]. As LN is also associated with *PIK3CA* mutations, it has been suggested they be included in the PROS group (see above) [3,63].

## 4. Liposarcoma

Liposarcoma consists of a heterogeneous group of mesenchymal tumors with adipocytic differentiation, showing a variable biological behavior, ranging from locally aggressive to metastasizing [10].

The World Health Organization (WHO) recognizes five histologic subtypes: well-differentiated liposarcoma/atypical lipomatous tumor (WDLPS/ALT), dedifferentiated liposarcoma (DDLPS), myxoid liposarcoma (MLPS), pleomorphic liposarcoma (PLPS), and myxoid pleomorphic liposarcoma (MPLPS). MPLPS is the latest entity, recognized by the WHO since 2020. All liposarcomas clinically present as large (over 5 cm in diameter) deep-seated, painless soft tissue masses.

Liposarcomas represent less than 2% of all pediatric soft tissue malignancies and nearly 90% of pediatric liposarcomas occur in the second decade of life. There is a female predominance and over 70% are myxoid types [10,77,78,79,80].

Peng et al. found that the most significant difference between young and adult patients with liposarcoma is the distribution spectrum of the subtype. The most common subtype among adult patients is ALT/WDLPS/DDLPS, comprising 50–60% of all liposarcomas, followed by MLPS (20–30%), PLPS (<5%), and MPLPS (rarely seen). In their study, the incidence of the MLPS subtype was high among young individuals (69.6%). The incidence of the ALT/WDLPS/DDLPS subtypes in their study cohort was only 13.0% and PLPS comprised 4.3% of the cases (the lowest incidence in pediatric patients). Historical series showed similar findings [78,79,81,82].

In their literature review, Baday et al. found that approximately 70% of patients with MLPS and WDLPS had surgical excision as the only treatment. Over 50% of patients with PLPS were treated with surgery, as well as radiotherapy and chemotherapy (most commonly a doxorubicin-based treatment). They concluded that more data on the efficacy of different pharmaceuticals in treating liposarcomas is needed; however, the rarity of these tumors in the pediatric population poses a challenge to obtaining this data [80].

Most patients have an excellent overall prognosis following surgical excision with negative margins as a single-modality treatment. Stanelle et al. found that central location of the primary tumor, high tumor grade, and positive surgical margins are strongly correlated with poor survival in pediatric patients with liposarcoma. They found that the five-year survival for patients with negative surgical margins was 95%, in comparison to a five-year survival rate of 50% in patients with positive surgical margins [82].

Baday et al. observed that prognosis was particularly determined by the histologic subtype. Patients diagnosed with MLPS and WDLPS, receiving minimal systemic therapy, had favorable outcomes: none experienced disease relapse or progression. The two patients with PLPS however, both died after receiving neoadjuvant chemotherapy with subsequent excision of the tumor [80].

### 4.1. Well-Differentiated Liposaroma (WDLPS)/Atypical Lipomatous Tumor (ALT)

The terms “atypical lipomatous tumor” (ALT) and “well-differentiated liposarcoma” (WDLPS) are synonyms for morphologically and genetically identical lesions. They represent a locally aggressive mesenchymal neoplasm, composed either entirely or partly of an adipocytic proliferation demonstrating at least focal nuclear atypia in both adipocytes and stromal cells. The choice of terminology between ALT and WDLPS should be based on the principle of avoiding either inadequate or excessive treatment. There is no potential for metastasis unless dedifferentiation occurs, therefore justifying the term “atypical lipomatous tumor” for lesions arising at anatomical sites where complete surgical resection is curative, so as to avoid over-treatment. For lesions arising in anatomical sites which have shown greater potential for disease progression (such as the retroperitoneum, spermatic cord, and mediastinum) the use of the term “well-differentiated liposarcoma” is more appropriate [3,83].

The extremities, followed by the head and neck, trunk, mediastinum and retroperitoneum are most commonly involved [10,84,85]. Although WDLPS/ALT is the most frequent liposarcoma in adults, it is exceptionally uncommon in the pediatric population [84,85,86].

WDLPS/ALT is characterized by a supernumerary ring and giant marker chromosomes, typically as the sole change, or simultaneous with a few other numerical or structural abnormalities [87]. Both supernumerary rings and giant markers contain amplified sequences originating from the 12q14–q15 region, with *MDM2* (12q15) being the main driver gene [83]. Several other genes located in the 12q14–q15 region (including *TSPAN31*, *CDK4* and *FRS2* (12q15)) are frequently co-amplified with *MDM2*. Amplification of *MDM2* and/or *CDK4* is almost always present, with the exception of Li-Fraumeni-associated cases, which show a *TP53* mutation (see also Table 1) [85,88,89,90,91,92,93,94]. Macroscopically, WDLPS/ALT shows a large, well-circumscribed, lobulated mass. The consistency varies, ranging from firm gray areas to gelatinous areas, depending on the proportion of fibrous and myxoid components [83].

Morphologic features of pediatric WDLPS/ALT are identical to those seen in older patients, dividing WDLPS into three subtypes: adipocytic (lipoma-like), sclerosing and inflammatory type [10,88]:Adipocytic (lipoma-like) is the most frequent subtype. It is composed of mature adipocytes with substantial variation in cell size, as well as cytonuclear atypia in adipocytes and/or stromal spindle cells. MDM2 and CDK4 immunohistochemical expression are typical, though in some cases difficult to evaluate, making fluorescence in situ hybridization (FISH) a valid alternative [10].The sclerosing subtype is most often seen in cases located in the retroperitoneum or spermatic cord. Scattered, bizarre stromal cells with marked nuclear hyperchromasia are seen, set in an extensive fibrillary collagenous stroma. The fibrous component may overshadow lipogenic areas, making it easy to miss in a small (biopsy) sample [10].The inflammatory type is the rarest subtype, occurring most often in the retroperitoneum and paratestis [3]. There is a predominant chronic inflammatory infiltrate, sometimes obscuring the adipocytic nature of the lesion [95,96].

WDLPS/ALT demonstrates an overall indolent course in young patients [81,84]. The significance of dedifferentiation in pediatric WDLPS is not known, due to the rarity of this event [10].

The most important prognostic factor is anatomical location. ALTs do not recur after complete excision. WDLPSs tend to recur repeatedly. In WDLPS uncontrolled local effects or, less often, systemic spread as a result of dedifferentiation may subsequently lead to death [83]. MDM2 inhibitors (SAR405838 and Nutlin-3A) have demonstrated preclinical (in-vitro) activity in liposarcomas [97,98].

### 4.2. Dedifferentiated Liposarcoma (DDLPS)

In some cases, WDLPS shows progression to a (usually non-lipogenic) sarcoma of variable histological grade, hence termed dedifferentiated liposarcoma [83]. Pediatric DDLPS is even more uncommon than WDLPS. Waters et al. observed a male to female ratio of one-to-one [88].

The genetic profile of DDLPS overlaps with WDLPS, both characterized by consistent amplification of *MDM2* and *CDK4* (see also Table 1) [89,99].

The macroscopic appearance of pediatric DDLPS is identical to its adult counterpart, usually consisting of a large multinodular yellow mass, containing discrete, solid, often tan-gray non-lipomatous (dedifferentiated) areas. The transition between the lipomatous and dedifferentiated areas may sometimes be gradual [10,83]. The microscopic appearance of pediatric DDLPS is also identical to those seen in older patients [10,88]. The areas of transition from WDLPS to DDLPS are usually abrupt. The extent of dedifferentiation is variable. These areas, however, most frequently resemble undifferentiated pleomorphic sarcomas or intermediate- to high-grade myxofibrosarcomas [77,83,100,101]. DDLPS characteristically shows the expression of MDM2 and/or CDK4, just as in WDLPS. Peng et al. found positivity for both CD34 and STAT6 in varying degrees. Although they found that STAT6 can be expressed in 7.4–14% of WDLPS/DDLPS, it is usually focally positive in contrast with the diffuse and nuclear pattern in solitary fibrous tumors [77].

The exact prognosis is not known due to the rarity of DDLPS in children and adolescents. However, the patient diagnosed with DDLPS in the study of Baday et al. died, despite aggressive, multimodal therapy [80].

### 4.3. Myxoid Liposarcoma (MLPS)

Myxoid liposarcoma (MLPS) is the most common liposarcoma subtype in pediatric patients. The lesions usually involve the extremities, trunk, head and neck, and abdominal regions [10,77,78,79,80,81]. There is no gender predilection and metastatic disease at initial presentation is uncommon [3,33].

MLPS has an identical macroscopic appearance to that of the respective tumors in adults: lesions are typically larger than 10 cm, circumscribed, multinodular and intramuscularly located (see also Figure 2). Higher grade tumors show a firmer, fleshy tan surface as opposed to a smooth, gelatinous and glistening surface in tumors of a lower grade [10,83].

Histological features of MLPS in children are also identical to those in adult cases. It presents as a moderately cellular, lobulated tumor with increased peripheral cellularity. There is a mixture of patternless, round to spindle-shaped non-lipogenic cells with variable numbers of small lipoblasts. These are set in a myxoid stroma with arborizing “chicken-wire” vessels. The spindle cells typically lack atypia and significant mitotic activity [83]. High-grade MLPS have a cellular overlap of more than five percent, showing diminished myxoid matrix, less apparent capillary vasculature, a higher nuclear grade, as well as increased mitotic activity. A corded/trabecular pattern is often present (see Figure 3) [33,83].

Immunohistochemical stains play little role in the diagnosis of MLPS but can be useful in the distinction of high-grade tumors from other round cell sarcomas [83]. There is no immunohistochemical expression of CDK4 or MDM2 [77]. Peng et al. found that 69.2% (9/13 cases) exhibited varying degrees of S100 protein positivity [77]. Both MLPS and high-grade MLPS harbor the same genetic abnormality [83]. Most cases are characterized by the t(12;16)(q13;p11) translocation, generating a *FUS-DDIT3* fusion transcript, translated into a chimeric oncoprotein that alters transcription and differentiation (see also Table 1) [83]. Peng et al. found *DDIT3* gene rearrangements in all 13 studied pediatric MLPS cases [77]. *DDIT3* rearrangement and the absence of *PLAG1* rearrangement help in the differential diagnosis of MLPS from extensively myxoid lipoblastoma [42,102].

Conventional MLPS in pediatric patients has an excellent prognosis [79]. Peng et al. observed disease-free survival in all their studied MLPS cases (13 cases in total) [77]. Progression to high-grade MLPS (formerly known as round cell liposarcoma) is extremely rare in children, and not clearly associated with a worse prognosis (likely owing to the rarity of these tumors) [10,103].

### 4.4. Pleomorphic Liposarcoma (PLPS)

Pleomorphic liposarcoma is a rare subtype of liposarcoma, with a low incidence in the pediatric population [77,83]. It presents as a rapidly growing mass, with a predilection for the extremities (primarily the upper limbs), frequently localized in the deep soft tissues. Some lesions occur at subcutaneous sites and rarely at dermal sites. Other anatomical sites, including trunk, retroperitoneum, head and neck, abdomen/pelvis, and cervical cord are less frequently affected [104].

Macroscopically, PLPS occurs as a large, well-circumscribed, non-encapsulated soft and multinodular lesion with a gelatinous appearance [83,104].

Histologic features are similar to their adult counterparts [77]. It is a high-grade lesion with pleomorphic features and a variable amount of lipoblastic differentiation [105]. The presence of pleomorphic lipoblasts (with clear intracytoplasmic vacuoles) is imperative for the diagnosis (Figure 4) [105]. Three histologic patterns have been noted [105]:

Pleomorphic/spindle cell areas with “malignant fibrous histiocytoma-like” appearance with scattered lipoblasts. This is the most common pattern, found in approximately two-thirds of the cases.Almost half of cases contain at least focal areas with intermediate- to high-grade myxofibrosarcoma-like morphology, with thick curvilinear vessels and cytonuclear atypia in myxoid areas, associated with pleomorphic lipoblasts [105,106].Approximately one quarter of the cases show an epithelioid morphology with scattered lipoblasts.

Within small biopsies, the diagnosis can be difficult, especially in cases with limited lipogenic differentiation [105]. Immunohistochemical staining for MDM2 and CDK4 is typically negative [11,104].

Molecular findings are in line with those of the adult PLPS and are associated with a complex karyotype with multiple (whole chromosomal) gains and losses, with the most frequent mutations being in *TP53* and *NF1* (see also Table 1) [77,105,107,108,109]. Peng et al. found loss of chromosomes 17 and 22, with deletion of *RB1* [77].

Although slightly better than in adults, the outcome is poor [77,78]. PLPS shows aggressive behavior with local recurrences and distant metastases to the lung and pleura. The overall five-year survival is 60%. Larger size and increased depth of invasion, localization and high mitotic activity are all related to a worse prognosis [104]. Local excision is the preferred treatment. Recurrence is, however, frequent despite aggressive multimodal therapy [78,80].

### 4.5. Myxoid Pleomorphic Liposarcoma (MPLPS)

MPLPS is a new entity within pediatric liposarcomas, first described in 2009 by Allagio et al. and genomically seen as a distinct subtype of liposarcoma in the current WHO (5th edition, 2020) [79,83,93,107]. These lesions are rare, although they have a higher incidence in the pediatric population in comparison to adults [77]. There is a female predominance, with a median age at diagnosis of 17.5 years. These lesions have a predilection for axial locations with a preference for the mediastinum [33,77,79,91]. In MPLPS occurring in a young patient and at an unusual anatomical location, genetic screening for Li-Fraumeni syndrome (LFS) is advised, since associations have been described [91,93,94].

MPLPS occurs as a large, non-encapsulated, deep-seated soft tissue lesion with ill-defined margins [107]. It is typically a multinodular growth with infiltration into the surrounding soft tissue [107].

Histologically, MPLPS is a high-grade lesion with areas of pleomorphic sarcoma (with severe cytological atypia, pleomorphic lipoblasts, brisk mitotic activity and occasional necrosis) combined with low-grade myxoid components (abundant myxoid matrix with bland, round to oval cells, scattered lipoblasts and a curvilinear, to plexiform capillary network) (Figure 5) [77,79,107]. Differentiation from PLPS can be challenging on small biopsies where the MLPS-like pattern is not prominent [77].

Immunohistochemically, MPLPS lacks expression of MDM2 and shows a loss of expression for RB1 [107]. MPLPS does not show a *DDIT3*-rearrangement, nor does it have an *MDM2* amplification, therefore, differing from MLPS and WDLPS/ALT, respectively [77,91].

The molecular features of MPLPS tend to show overlap with PLPS, with inactivation of *RB1* and a complex chromosomal profile (hyperdiploid/hypotriploid karyotype) with gains and losses of chromosomes with deletions/mutations of *TP53,* and deletions of *KMT2D* or *NF1* (see also Table 1) [33,77,91,105,106,107,109,110]. However, there are typically focal copy number changes in MPLPS as opposed to large/whole chromosomal gains and losses in PLPS [107,109]. Gains are also more frequent in PLPS, whereas losses predominate in MPLPS [106]. Association with LFS (caused by *TP53* germline mutations) is also known [33,77,91,110].

MPLPS is associated with a poor prognosis, with local recurrences and distant metastases to bone, lung and soft tissue. MPLPS has the highest mortality rate of all pediatric liposarcomas [77,91,107].

Wide local excision is preferred, with 1.5 cm margins. Adjuvant radiotherapy after surgical excision is advised in cases of microscopically incomplete resection, or if surgical margins are marginal [91]. The role of chemotherapy is controversial. It is sometimes used in combination with radiotherapy for unresectable lesions [91]. The use of doxorubicin has been reported in the treatment of MPLPS [110].

## Figures and Tables

**Figure 1 jcm-11-01938-f001:**
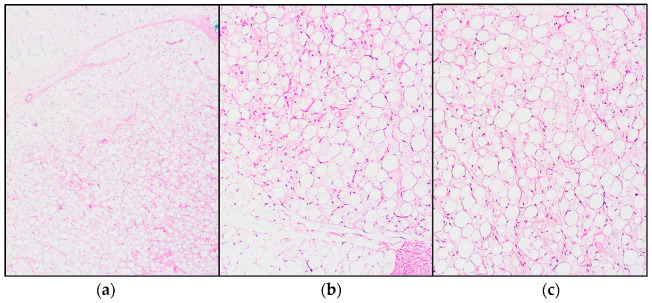
Histopathology of a lipoblastoma: (**a**) Overview showing lobulated sheets of adipocytes, separated by a fibrovascular septum (hematoxylin and eosin (H&E) staining); (**b**,**c**) Close up showing an image of the adipocytes with a spectrum of maturation, ranging from spindled mesenchymal cells to lipoblasts to mature adipocytes (H&E staining).

**Figure 2 jcm-11-01938-f002:**
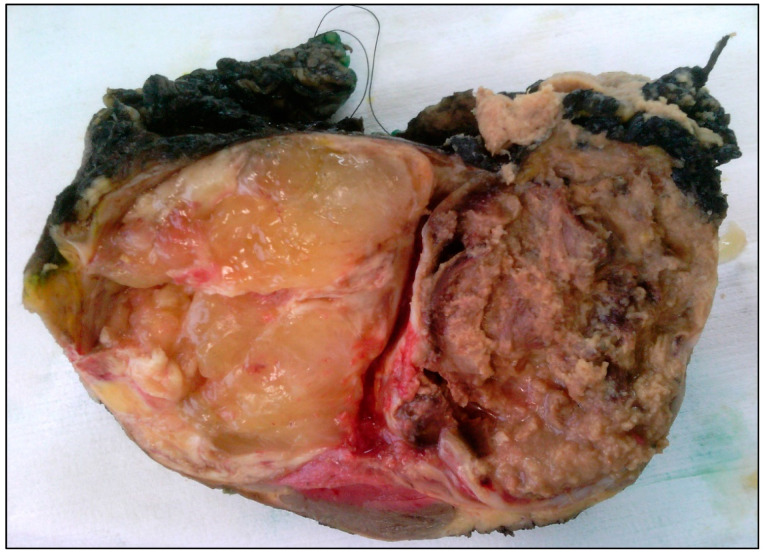
Macroscopic image of a pediatric low-grade myxoid liposarcoma, excised after neoadjuvant radiotherapy. On the left side of the specimen is a smooth, gelatinous area (characteristically seen in tumors of a lower grade). The right side of the specimen shows necrosis, as a result of the preoperative radiotherapy.

**Figure 3 jcm-11-01938-f003:**
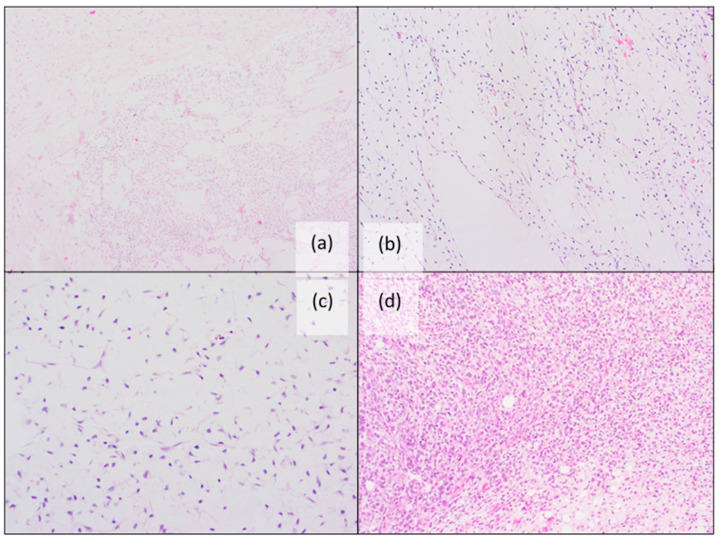
Histopathological features of myxoid liposarcoma: (**a**) Characteristic alveolar and edema-like growth pattern (H&E staining); (**b**,**c**) Myxoid stroma with arborizing “chicken-wire” vessels and lipoblasts (H&E staining); (**d**) High-grade MLPS with more than 5% cellular overlap, diminished myxoid matrix, less apparent capillary vasculature, a higher nuclear grade and increased mitotic activity (H&E staining).

**Figure 4 jcm-11-01938-f004:**
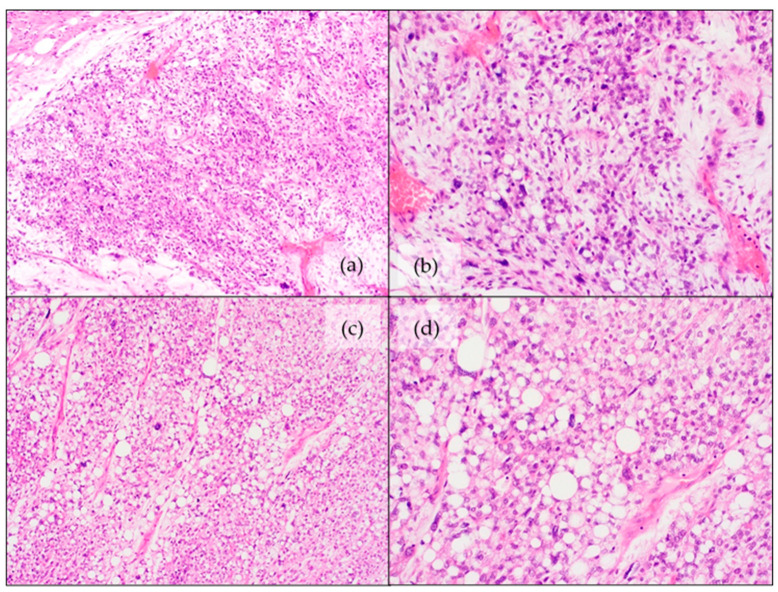
Histomorphological features of pleomorphic liposarcoma: (**a**,**b**) Overview and detail images with sporadic, pleomorphic lipoblasts (H&E staining); (**c**,**d**) Overview image and magnified displaying more pronounced pleomorphic lipoblasts (H&E staining).

**Figure 5 jcm-11-01938-f005:**
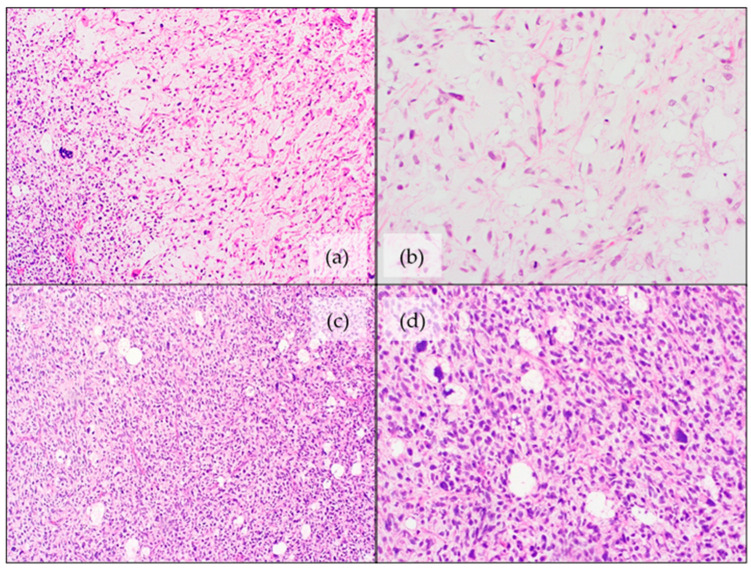
Histopathological features of myxoid pleomorphic liposarcoma: (**a**) Transition from a myxoid liposarcoma-like area to a pleomorphic liposarcoma-like area (H&E staining); (**b**) Detail image of myxoid liposarcoma-like zone with “chicken-wire” vessels and lipoblasts. There is “at random” atypia and increased mitotic activity (H&E staining); (**c**,**d**) Overview image and detailed image of pleomorphic liposarcoma-like area with severe cytonuclear atypia and pleomorphic lipoblasts (H&E staining).

**Table 1 jcm-11-01938-t001:** WHO classification of pediatric adipocytic tumors and the most frequently associated cytogenetic/molecular characteristics.

WHO Classification of Pediatric Adipocytic Neoplasms	Most Frequently Associated Cytogenetic/Molecular Characteristics
Lipoblastoma/lipoblastomatosis	Structural alteration of chromosome 8q leading to *PLAG1* rearrangements
Lipomatosis	Germline, mosaic or somatic mutations of *PTEN*, *PIK3CA* and *TSC*
Well differentiated liposarcoma/Atypical lipomatous tumor	Amplification of *MDM2* and/or *CDK4* Except in Li-Fraumeni-associated cases: *TP53* germline mutation
Dedifferentiated liposarcoma	
Myxoid liposarcoma	t(12;16)(q13;p11) translocation, generating *FUS-DDIT3* fusion transcripts
Pleomorphic liposarcoma	Complex karyotype with multiple (whole chromosomal) gains and losses, most common mutations in *TP53* and *NF1*
Myxoid pleomorphic liposarcoma	Inactivation of *RB1* and a complex chromosomal profile with gains and losses of chromosomes with deletions/mutations of *TP53*, and deletions of *KMT2D* or *NF1*

## Data Availability

Not applicable.

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
