# Peer review of "Update of Pediatric Lipomatous Lesions: A Clinicopathological, Immunohistochemical and Molecular Overview"

_jcm, 2022, doi:10.3390/jcm11071938_

Round 1

Reviewer 1 Report

This review article on adipocytic tumors in the pediatric population is well-written and gives a good overall coverage of the field. I miss representative radiological images/clinical images of these tumors. I would further welcome a more detailed review of the oncological treatment of liposarcomas. 

Author Response

Thank you very much for reviewing our article.

Point 1: I miss representative radiological images/clinical images of these tumors.
Response 1: A macroscopic image was added to the manuscript (see figure 2).

Point 2: I would further welcome a more detailed review of the oncological treatment of liposarcomas.
Response 2: Given the rarity of pediatric liposarcomas, data is limited. Literature findings on the general treatment of liposarcomas (including chemotherapy) were added to the manuscript.

Reviewer 2 Report

The articles is a review of paediatric lipomatous lesions from a pathoclinical perspective.

Grammar/ Language:

  • A high quality of English is present. One remark is that the authors steer clear of starting a sentence with a number. One example is "75% to 90%...". The reviewer suggests starting with "In 75% to..."

Context:

  • The delineation of the subtypes are describe in a consice yet relevant manner
  • When the author mention survival, clinical paediatric oncologists would prefer survival contextualised in percentages rather than a non-defined statement such as "... survival is excellent..."
  • In the introduction the authors state that adult and paeditric lipomatous lesions differ yet the statement is never qualified in the text. The reviewer suggests some contexualisation of the statement or eluding in the discription of the subtypes that it is a unique paediatric feature or not.

Author Response

Thank you very much for reviewing our article.

Point 1: One remark is that the authors steer clear of starting a sentence with a number. One example is "75% to 90%...". The reviewer suggests starting with "In 75% to..."
Response 1: This sentence has been altered so it no longer starts with a number.

Point 2: When the author mention survival, clinical paediatric oncologists would prefer survival contextualised in percentages rather than a non-defined statement such as "... survival is excellent..."
Response 2: Results of different studies have been incorporated into the manuscript to contextualize the prognostic statements made.

Point 3: In the introduction the authors state that adult and paeditric lipomatous lesions differ yet the statement is never qualified in the text. The reviewer suggests some contexualisation of the statement or eluding in the discription of the subtypes that it is a unique paediatric feature or not.
Response 3: The most apparent difference between adult and pediatric lipomatous lesions is with respect to the distribution spectrum of the (sub)types. In adults well-differentiated liposarcoma / atypical lipomatous tumor (WDLPS/ALT) is the most common liposarcoma. In the pediatric population WDLPS/ALT is exceptionally rare, and the most common liposarcoma in children is myxoid liposarcoma (ML). Myxoid pleomorphic liposarcoma (MPLPS) is rare, although there is a higher incidence in the pediatric population in comparison to adults. The additional information was included in the manuscript.

Reviewer 3 Report

Well-written summation of pediatric lipomatous tumors. Please also incorporate findings of and cite: https://pubmed.ncbi.nlm.nih.gov/31282548/ 

Author Response

Point 1: Please also incorporate findings of and cite: https://pubmed.ncbi.nlm.nih.gov/31282548/
Response 1: Thank you very much for reviewing our article. The findings of the article mentioned above have been incorporated in our manuscript and the article has been cited.